# Preoperative Perforator Mapping in DIEP Flaps for Breast Reconstruction. The Impact of New Contrast-Enhanced Ultrasound Techniques

**DOI:** 10.3390/jpm13010064

**Published:** 2022-12-28

**Authors:** Max J Zinser, Nadja Kröger, Wolfram Malter, Tino Schulz, Michael Puesken, Peter Mallmann, Matthias Zirk, Kai Schröder, Christoph Andree, Kathrin Seidenstuecker, David Maintz, Ralf Smeets, Christian Eichler, Oliver C Thamm, Carola Heneweer

**Affiliations:** 1Division of Plastic-, Reconstructive- and Aesthetic Surgery, University Hospital Cologne, Kerpener Str. 32, 50937 Cologne, Germany; 2Breast Center, Department of Obstetrics and Gynecology, University Hospital Cologne, Kerpener Str. 32, 50937 Cologne, Germany; 3Department of Plastic Surgery, Sana-Clinic, Gräulinger Str. 120, 40625 Düsseldorf-Gerresheim, Germany; 4Institute for Diagnostic and Interventional Radiology, University Hospital Cologne, Kerpener Str. 32, 50937 Cologne, Germany; 5Department of Plastic Surgery, Sana-Clinic, Urdenbacher Allee 83, 40593 Düsseldorf-Benrath, Germany; 6Division of Regenerative Orofacial Medicine, University Medical Center Hamburg-Eppendorf, Martinistr. 52, 20251 Hamburg, Germany; 7Breast Center, St. Franziskus-Hospital, Hohenzollernring 72, 48145 Münster, Germany; 8Department of Plastic Surgery, University Witten-Herdecke, Helios Hospital Berlin-Buch, Schwanebecker Chaussee 50, 13125 Berlin, Germany

**Keywords:** DIEP flap, autologous breast reconstruction, perforator mapping, contrast-enhanced ultrasound (CEUS), CT angiography (CTA), B-flow imaging

## Abstract

Deep inferior epigastric artery flaps (DIEP) represent the gold standard of autologous breast reconstruction. Due to significant variations in vascular anatomy, preoperative perforator mapping (PM) is mandatory in order to ensure the presence of a sufficient perforator within the flap. In this regard, CT angiography (CTA) is currently the method of choice. Therefore, we investigated the value of contrast-enhanced ultrasound (CEUS) techniques for preoperative PM in comparison to CTA. Patients underwent PM, utilizing both CTA and CEUS techniques. Documentation included the course of the vascular pedicle through the rectus muscle (M), fascial penetration (F), the subcutaneous plexus (P) and the skin point (SP) on either side of the abdomen. Thus, contrast-enhanced B-Flow (BCEUS), B-Flow ultrasound (BUS), CEUS, color Doppler ultrasound (CDUS) and CTA were evaluated in terms of the diagnostic consistency and effectiveness of PM. Precision (∆L) was then calculated in relation to the actual intraoperative location. Statistical analysis included Kruskall–Wallis, Levene and Bonferroni tests, as well as Spearman correlations. A total of 39 DIEP flaps were analyzed. Only CTA (∆L = 2.85 mm) and BCEUS (∆L = 4.57 mm) enabled complete PM, also including P and SP, whereas CDUS, CEUS and BUS enabled clear PM throughout M and F only. Regarding the number of detected perforators, PM techniques are ranked from high to low as follows: CTA, BCEUS, BUS, CEUS and CDUS. CTA and BCEUS showed sufficient diagnostic consistency for SP, P and F, while CDUS and CTA had a superior performance for M. BCEUS offers precise image-controlled surface tags and dynamic information for PM without imposing radiation and may, therefore, be considered a feasible add-on or alternative to CTA. However, BCEUS requires an experienced examiner and is more time-consuming.

## 1. Introduction

Microvascular free tissue transfer for breast reconstruction was introduced 30 years ago by Holmstroem [1], Robbins [2] and Hartrampf [3], including various abdominal donor site flaps, such as the transverse rectus abdominis muscle flap (TRAM) and the deep inferior epigastric perforator flap (DIEP). In particular, the development of the DIEP flap for autologous breast reconstruction has been a major surgical advance, as it now represents the first choice of autologous breast reconstruction [4] due to low donor site morbidity, excellent aesthetic results [5]—such as an improved abdominal contour with diminished adverse outcomes and a relatively short hospital stay compared to other flap reconstructions, such as the TRAM [3]—and the superior gluteal artery perforator flap.

However, preoperative DIEP-PM remains mandatory in the setting of presurgical planning. The vascular anatomy of the DIEP vessels exhibits a large variability regarding quantity, location and diameter within a single patient and also differs widely when comparing one patient to another [6]. In addition, the survival of a perforator flap relies grossly on the number of perforators and their central positioning within the transferred tissue, hereby enabling adequate blood supply. There are various methods of PM [7] described in the literature, including, among others, color duplex [8,9], fluorescent angiography, infrared thermography, CTA [8,10,11] and magnetic resonance angiography (MRA) [7,12]. A multidisciplinary consensus study [13] defined CTA as the gold standard for PM imaging due to the fact that it allows precise anatomical description of perforator origin, including the intramuscular course (M), fascial penetration (F), the subcutaneous plexus (P) and the skin point (SP). However, the main disadvantages of CTA comprise the exposure to radiation, potential adverse effects in relation to the contrast medium and the lack of dynamic information (e.g., velocity, flow). Additionally, CTA does not allow for clinical surface navigation, meaning all landmarks (SP, F, M) must be related to the umbilicus (Figure 1). Due to the mobility of soft tissue, this method may be imprecise. In contrast, Doppler flowmetry and CDUS provide the surgeon with prior anatomical surface location of the perforator without exposure to radiation. In addition, CDUS may offer dynamic information regarding perforator perfusion, but it is not further capable of precisely determining P and SP with clear delineation [14,15,16,17,18]. This is attributed to strict angular dependence, low frame rates and reduced resolution. However, literature remains controversial, as Blondeel [9] reported of a sensitivity of 96.2% for PM in DIEP surgery via CDUS. In contrast, Rozen [15] compared CDUS with CT angiography and reported a complete failure of perforator detection via CDUS, while Scott [19] reported a sensitivity of 66.3%. Consequently, CEUS was introduced by Su [14] as a new technique for PM, as ultrasound contrast agents provided additional reflectors, and thus increased the sensitivity of CEUS-PM. Although CEUS may successfully delineate the vascular course, precise navigation of P, including SP remains challenging. In order to counterfeit this, DIEP patients were preoperatively subjected to BCEUS imaging for PM. Th PM of small vessels, including P and SP should thus become possible when utilizing BUS paired with contrast agents. Additionally, artifacts, such as aliasing or blooming, may possibly be avoided. Preliminary data from our previously published study [17] clearly showed the value of BCEUS for PM of anterio-lateral thigh flaps.

To our knowledge, no study has so far explicitly analyzed the feasibility, diagnostic effectiveness and precision of contrast-enhanced ultrasound (BCEUS, CEUS, BUS and CDUS) in comparison to CTA for DIEP-PM.

Thus, we hypothesized that an algorithm combining CDUS, for the identification of the inferior-epigastric vessels and M, with BCEUS in order to locate F, P and SP may function as a feasible supplement or even as an alternative to CTA in preoperative DIEP PM, while simultaneously circumventing radiation and enabling precise PM, including direct surface navigation.

## 2. Materials and Methods

Autologous breast reconstruction was performed via 39 DIEP flaps. All patients received preoperative CTA and an ultrasound examination for perforator mapping according to best clinical care. Informed consent was obtained, and the study was approved from the university’s ethical committee. The study was performed according to the Helsinki protocol. Patients with an intolerance for sulfur-hexafluoride, a history of cardiovascular disease or pulmonary hypertonia were excluded. Clinical patient data, including age, body-mass index and type of breast reconstruction (single vs. double), are summarized in Table 1. Furthermore, the anatomical DIEP branching pattern, according to Moon and Taylor (type I-III) [20], was included. Ultrasound examination (CDUS, CEUS, BUS and BCEUS) was performed by a single sonographer and CTA was performed according to our standard protocol (Table 2).

### 2.1. Ultrasound Examinations

Preoperative ultrasound was performed with a LOGIQ E9 scanner (GE Healthcare) and equipped with a linear [2–8 MHz] and matrix probe [4–15 MHz] [17,21], respectively.

Bilateral abdominal wall duplex was performed on all patients, according to the following sequence (Table 2):A 2–8 MHz transducer was utilized with settings aligned to the depiction of peripheral arteries. First, large vessels, including the external and internal iliac arteries, were identified, followed by the identification of the origin of the deep inferior epigastric arteries (DIEA).Settings were then adjusted in order to visualize lower-flow arterial vessels. Complete PM then included the following:
(a)The muscular course (M) of the vessels was displayed according to the intramuscular branching pattern [20] (Table 1 and Table 3).(b)F was then analyzed, including P and SP (Appendix A, Figure 2). PM was performed with each technique, namely CDUS, CEUS, BUS and BCEUS and was subsequently documented and recorded by cine loops and freeze frames and traced to the skin (Table 1, Table 2, Table 3, Table 4, Table 5 and Table 6, Figure 2, Figure 3 and Figure 4). For CEUS and BCEUS, the contrast agent SonoVue (Bracco^®^) was prepared according to the guidelines provided [21,22]. The contrast agent was fully dissolved in 15 mL of saline. An amount of 2 mL was injected intravenously, followed by a bolus of 10 mL saline. The blood flow signal was enhanced without excessive overflow [22]. It was then evaluated whether the course of F, P and SP could be visualized with clear delineation (Figure 2 and Figure 4).

### 2.2. Computed Tomographic Angiography

All patients underwent CTA, according to our standard protocol (Table 2). After intravenous administration of iodinated contrast agent followed by a 30 mL saline chaser, scans were started with a delay of 30 s after passing the predetermined threshold of 150 HU within the abdominal aorta. Locations of SP, F and M were measured by distance from the umbilicus for presurgical planning and intraoperative transformation (Figure 1, Figure 2 and Figure 4). Perforators were transposed on dedicated 3D volume renderings showing the ventral abdomen.

### 2.3. Comparison Ultrasound Techniques versus CT-Angiography

The main focus of this study was the comparison of ultrasound techniques versus CTA. PM for each technique was examined and evaluated as follows:Precision of perforator mapping.Number of perforators detected at the fascial level (F).Delineation and diagnostic efficiency of the ultrasound modes compared to CTA.

#### 2.3.1. Precision of Perforator Mapping

Preferred perforators were selected according to their internal diameter, blood flow pattern and central position within the DIEP flap and subsequently marked on each hemi-abdomen, as recorded by the different ultrasound techniques (Figure 4). The chosen SP was determined to lie within a range of 4 cm above and 10 cm below the umbilicus vertically and exhibit a diameter of >1 mm at F. For CTA, perforator mapping, including M, F, and SP, was displayed in the axial, transverse and sagittal views, as well as in the 3D image (Figure 1). Since CTA does not allow direct point-to-patient navigation, the landmarks (M, F and SP) have to be determined in relation to a fixpoint (umbilicus). Thus, the umbilicus was displayed in 3D and the respective distances from M, F and SP were measured preoperatively within the CTA data set, and PM was then navigated intraoperatively. In contrast, the ultrasound techniques (CDUS, BUS, CEUS and BCEUS) allow preoperative PM (DIEA, M, F, SP) directly on the skin of the abdomen (Figure 4).

The precision of PM was evaluated for each technique (Table 3, Figure 3). Therefore, the landmarks of PM, SP, F and M, were then compared with the intraoperative situation (Table 3, Figure 3 and Figure 4). Metric deviation (∆L mm) was then analyzed.

The final decision of perforator selection was nevertheless made intraoperatively following clinical evaluation of the perforators and utilizing the clamping test (Figure 4). When flap perfusion (e.g., venous congestion) remained unaffected during clamping, the corresponding perforator was chosen. Otherwise, additional perforators were considered.

#### 2.3.2. Number of Perforators Detected at the Fascial Level (F)

For each diagnostic technique (CTA, CDUS, CEUS, BUS, BCEUS), the sum of the perforators at the fascial penetration point (F) was determined and compared with the actual intraoperative situation. The Spearman correlation (R) was then calculated.

#### 2.3.3. Delineation and Diagnostic Efficiency Compared to CTA

CTA enables a fast and detailed 3D view of PM, including DIEA, M, F, P and SP. Thus, CTA was used as a reference. The delineation of PM was compared to CTA, and the diagnostic efficiency was calculated via sensitivity, specificity and Youden index.

The final decision of perforator selection was nevertheless made intraoperatively following clinical evaluation of the perfusion of the flap and utilizing the clamping test (Figure 4). When flap perfusion (e.g., venous congestion) remained unaffected during clamping, the corresponding perforator was chosen. Otherwise additional perforators were considered.

### 2.4. Statistical Analysis

Statistical analysis was performed with SPSS 27.0 (IBM, Armonk, NY, USA). Acquired data followed normal distribution. Mean values (∆L), including standard deviations and confidence intervals, which were set at 95%, were calculated. The accuracy of perforator mapping for each technique was analyzed by calculating (∆L) between preoperative imaging and the actual assessed vascular course during surgery via Kruskal–Wallis, Levene and Bonferroni posthoc tests for multiple comparisons. Correlation was calculated using the Spearman coefficient. Diagnostic consistency and efficacy were assessed via sensitivity, specificity and Youden index.

## 3. Results

A total of 39 breast reconstructions were performed while 9 patients underwent double-sided DIEP flap reconstruction and 21 patients unilateral reconstruction. The mean age was 52 yrs, and the mean BMI was 24.7 (Table 1).

### 3.1. Precision of PM

According to our findings, we recommend the following algorithm (Figure 2 and Figure 4, Table 3).

The DIEA and its muscular course (M) should be mapped via CDUS, since CDUS was the most precise technique (∆L = 4.77 mm ± 0.86). Additionally, the vascular course through M was easily displayed, utilizing CDUS, and accuracy was superior compared to the other techniques (∆L = 5.92 mm ± 1.32), while BCEUS exhibited the highest precision (∆L = 3.63 mm ± 0.84) in terms of displaying F, a very relevant structure for flap design. BUS showed reasonable precision only when BMI was low, ultimately resulting in overall unconcise results. P and SP remain the most important landmarks in preoperative PM and should thus be located within the center of the flap.

Ultimately, a clear display of SP was only feasible via CTA (∆L = 2.85 mm ± 1.26) and BCEUS (Appendix A) (∆L = 4.57 mm ± 1.34). Clear signals were neither detectable via CDUS (Appendix A) nor BUS. CEUS was grossly inaccurate (∆L = 8.00 mm ± 0.82) and displayed only inconsistent delineation.

#### 3.1.1. Number of Perforators Detected with Each Technique at the Fascial Level (F)

Table 4 displays the comparison of the number of perforators, which were detected on the epifascial level (F) via preoperative PM, utilizing either CTA, CDUS, CEUS, BUS or BCEUS with the actual number of perforators detected intraoperatively (Figure 1, Figure 2 and Figure 4). Results for CTA (r = 0.98) and BCEUS (r = 0.95) were superior, although no significant difference could be detected, whereas CDUS (r = 0.091), CEUS (r = 0.225) and BUS (r = 0.584) were only able to identify a significantly lower amount of perforators at the epifascial level F. This indicates that CTA remains advantageous for PM, nevertheless acknowledging that the BCEUS technique using contrast-enhanced B-mode (r = 0.95) was the only feasible alternative for PM at level F (r = 0.95).

#### 3.1.2. Delineation and Diagnostic Strength of the Ultrasound Modes

CTA should be considered the gold standard for PM as it enables the complete and fast visualization of vessels with defined delineation, also including a 3D view.

We thus compared the different ultrasound modes with CTA (Table 5) and analyzed their diagnostic efficiency. Therefore, sensitivity, specificity and Youden index were calculated. CDUS was the best alternative to CTA (YI = 86.7) regarding the display of the DIEA and M. In contrast, enhanced ultrasound techniques were necessary to visualize F. BCEUS (YI = 92.9) and CEUS (YI = 82.9) were thus feasible alternatives to CTA. Tremendous differences became evident when PM distal to F was demanded. For P and SP, only BCEUS enabled comparable visualization to CTA (YI = 83.3). Here, reasonable visualization for SP was only possible via BCEUS, whereas the other ultrasound modes were not capable of concisely displaying SP with clear delineation (Table 5).

Summarizing the results of this study, we now recommend the following algorithm for PM utilizing ultrasound modes as an alternative to CTA (Table 6).

PM should be started at the DIEA (origin). Then, M should be displayed. Therefore, we recommend CDUS, since contrast-enhanced techniques (BCEUS and CEUS) were not necessary for better diagnostic efficiency (Table 5).BCEUS should then be utilized to visualize F, P and SP (Appendix A). The contrast-enhanced technique BCEUS is required to obtain clear delineation for F, P and SP (Table 5).

However, the mean time of ultrasound PM was 45 min (±12 min), and thus significantly longer than CTA 12 min (±2 min). No complete flap losses were recorded. A total of 2 patients experienced abdominal seroma, and 2 others needed surgical revision due to partial flap necrosis.

## 4. Discussion

Because of their superior aesthetic results and low morbidity rates [5], DIEP flaps belong to the preferred methods of autologous breast reconstruction. There lies consensus within the literature that precise preoperative PM is mandatory to increasing safety and reducing surgical time [23,24,25]. The main goal of this study was to analyze whether the ultrasound modes, CDUS and BUS, or new contrast-enhanced ultrasound techniques, such as BCEUS and CEUS, are feasible alternatives to CTA. Advantages include the absence of radiation and iodinated contrast agents and direct surface tagging of perforators (F and SP) on the patient, hereby enabling secure preoperative flap planning by ensuring a centrally located SP within the flap.

In our study, we, therefore, propose an alternative algorithm for PM, utilizing CDUS and BCEUS. With CDUS, there were no significant differences in comparison to CTA with regard to the visualization of the DIEA and M. For the display of F, P and SP, BCEUS was a reliable alternative to CTA without significant differences in terms of diagnostic precision and diagnostic efficiency. Additionally, cutaneous tagging and navigation via BCEUS was even more precise and advantageous compared to CTA for F due to real-time control during ultrasound.

In contrast, CTA allows for the easy and fast 3D mapping of all landmarks, but intraoperative transfer may be challenging, as it lacks direct surface mapping and landmarks must be measured in relation to the umbilicus (Figure 1) [15]. A consensus article considered CTA as the preferred method for PM and published literature provides evidence that sensitivity reaches 100% [13,15,23]. Up until now, there is only negligible and very controversial literature on the topic of CTA and ultrasound comparison. Blondeel [9] reported a sensitivity of more than 90% when utilizing CDUS for PM, but others could not affirm these findings. Giunta [26] reported a sensitivity of 89%, Scott [19] stated 65.3%, while Rozen [27] and Cina [11] merely reported that perforators could not be found via CDUS. Klasson [16] found no significant differences between handheld Doppler and CTA for PM. However, none of these studies clearly differentiated between F, P and SP. Therefore, it may be possible that SP was wrongly identified as P or even F. In contrast to the above-mentioned results, Mijuskovic [5] reported a higher sensitivity for PM via CDUS in comparison to CTA. Following our experience, we do not recommend the handheld Doppler due to its inability to visualize M, F and SP. Based on the results of this study, we recommend starting with CDUS to analyze the DIEA and M, before switching to BCEUS in order to display and determine F and SP.

According to Moon [20], CTA can accurately demonstrate branching patterns of the DIEA in 3D. This provides the surgeon with important information, since type 1 (one single common trunk) enables easy vascular dissection, whereas with type 2 or 3, the vascular course is less conductive to its perforating pattern and intramuscular dissection, and thus becomes more challenging [25].

Some authors [19] prefer PM via CTA due to its ability to perform 3D imaging of the inferior and superior (SIA) epigastric arteries. This additionally permits preoperative planning of an SIEA flap. However, all of this information may also be gathered via our proposed ultrasound algorithm while the presentation of 3D imaging was found to be more appropriate with CTA rather than ultrasound. This may be due to the fact that surgeons are not always well acquainted to the use of ultrasound PM. CTA, including 3D volume rendering, takes about 15 min, whereas contrast ultrasound PM takes markedly longer. In our study, the average examination time was 45 min, although other authors reported times of up to 2 h [28]. The 3D visualization of the perforator vessels (DIEP and SIEV) is fast, convenient and reliable as it enables the surgeon to plan the flap quickly. Perhaps this is another argument for why CTA remains the gold standard.

In our study, all BCEUS examinations were performed by a single highly specialized sonographer, which may have contributed to the high correlation with CTA. Intrinsic interobserver variability of the ultrasound might, therefore, be an additional handicap. Indeed, Scott [19] abandoned preoperative CDUS due to high false-negative rates. This phenomenon is referred to as “overflow” and is caused by an acceleration of blood flow signals due to the accumulation of contrast-enhanced bubbles. The backflow deriving from small veins may then be mistakenly misdiagnosed as small perforating arteries. This phenomenon was not observed in our study. B-mode was superior with regard to the display of the blood flow direction and was also able to detect flow origin. Furthermore, the B-flow technique [29] (BCEUS) is stated to be independent of the Doppler effect and relies on a subtraction algorithm, which then provides high spatial resolution equivalent to B-mode imaging. Therefore, BCEUS is advantageous compared to CEUS. A further rationale that supports CTA includes the ability to evaluate additional information by morphometric measurements, e.g., the patients’ individual risk of donor site morbidity [30,31] or breast cancer recurrences [32].

In conclusion, the various advantages of ultrasound techniques include, among others, the absence of radiation and precise surface navigation by tagging perforator positions cutaneously. Furthermore, BCEUS is capable of displaying dynamic vessel parameters (trajectory of blood flow) and may thus differentiate between arteries and veins. In an experienced setup, BCEUS may also be beneficial, since rising numbers of prophylactic mastectomies in BRCA carriers, which are of significantly younger age, call for radiation-free PM [33].

Consequently, Su [14] combined CEUS with CDUS. However, precise visualization of P and SP remained challenging. This may be attributed to unfavorable Doppler angles, since perforators frequently exhibit tortuosity. In our experience, vessels are often delineated irregularly and, in particular, smaller branches may not always be traceable to a certain feeder vessel. These difficulties may be overcome by utilizing either extremely sensitive Doppler methods, such as superb microvascular imaging (SMI, Canon), MicroFlow-Imaging (Philips) or BCEUS, as was the case in our study (Figure 2). Since BCEUS is based on a subtraction algorithm, obtained signals are independent of the angle between the vessel and the transducer. This leads to smooth vessel delineation, including even smaller branches. Although CTA and MRA are both suitable for PM [4,7,34,35,36], they do not provide hemodynamic information and precise surface mapping [34,35,36]. We, therefore, recommend BCEUS as a valuable supplement or even an alternative for PM, as SP and F can be visualized with clear delineation. However, further challenges remain: Three-dimensional stacks cannot be obtained with 4D transducers utilizing neither BUS nor BCEUS due to the large amount of data that require corresponding processing. As of now, only 3D sweeps can be performed. However, this problem may be solved soon with the aid of new rendering options. Four-dimensional BCEUS would thus be desirable. Additionally, other Doppler-based technologies, such as SMI, MicroFlow or optimized PowerDoppler settings, must also be tested and compared, since special modes deriving from a single vendor may not be optimal for broad implementation in a clinical routine. Further studies, including larger subject numbers, addressing the evaluation of valuable perforator hemodynamic parameters, such as peak velocity and resistance index for quality perforator assessment, are nevertheless required.

We found the algorithm combining CDUS and BCEUS to be an equivalent alternative to CTA for PM and will rely on CDUS/BCEUS ultrasound perforator mapping in the future.

## 5. Conclusions

Preoperative PM is mandatory for DIEP flap planning as it significantly reduces surgical time and improves flap harvest safety. We found that BCEUS combined with CDUS is a feasible supplement or may even function as an alternative to CTA. Surface navigation via BCEUS allows for precise PM with a sensitivity of 91.2%, a specificity of 88.9% and a Youden Index of 80.1% without concomitant radiation. However, the examination requires an experienced sonographer.

## Figures and Tables

**Figure 1 jpm-13-00064-f001:**
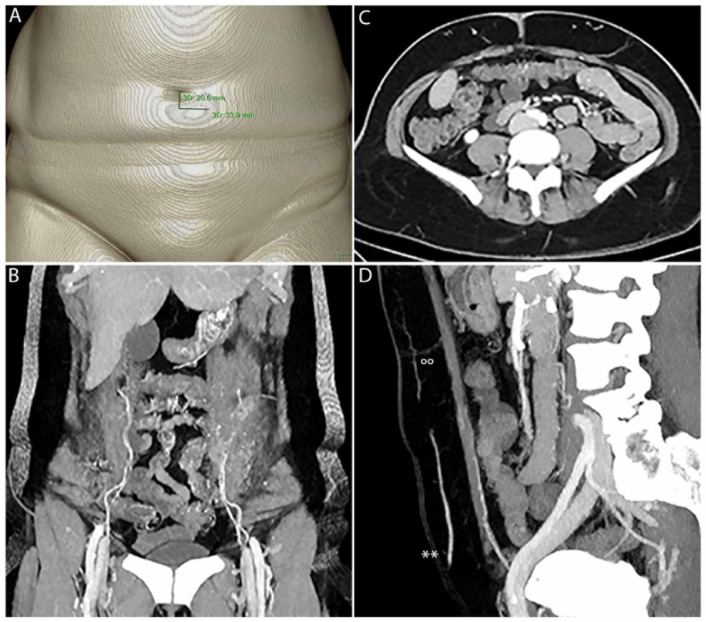
CTA enables fast 3D PM with clear delineation of SP, P, F, M and the IEA. SP must be measured manually in relation to the umbilicus (**A**). (**B**) shows PM in 3D. The origin of the IEA, including M with type II branching on the left and type I branching on the right, is clearly visible. In (**C**), F and P of the perforators on the left and right side are clearly visible. IEA, M, F, P(°°) and SP can be clearly seen within the sagittal view (**D**), including the superficial epigastric artery (SIEA**).

**Figure 2 jpm-13-00064-f002:**
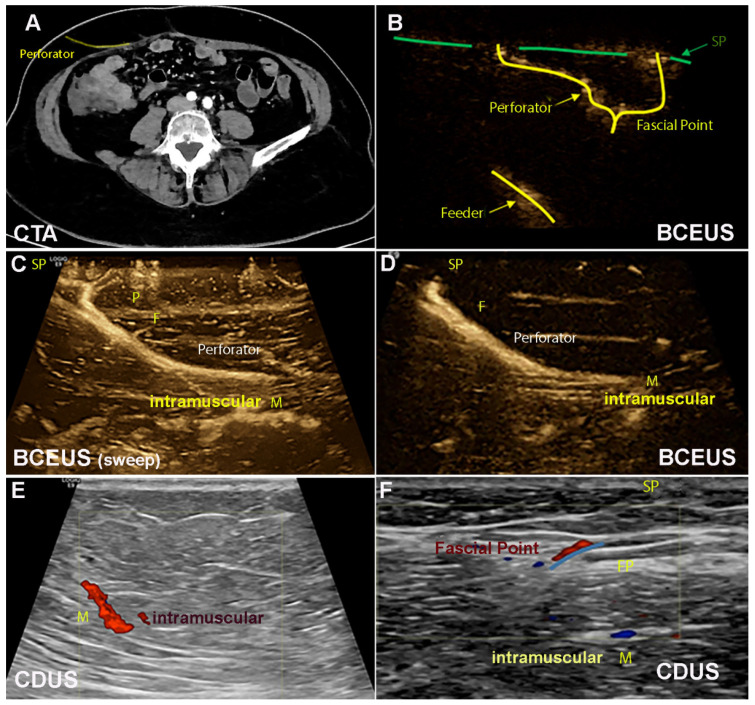
(**A**) illustrates the DIEP perforator in CTA axial view. (**B**–**F**) show PM using BCEUS and CDUS. (**B**) systematically illustrates the feeder F, P and SP. Only BCEUS is sufficiently able to display P and SP ((**C**) 3D sweep, (**D**)). Illustration of the IEA and M is easily feasible via CDUS (**E**,**F**). However, P and SP cannot be displayed.

**Figure 3 jpm-13-00064-f003:**
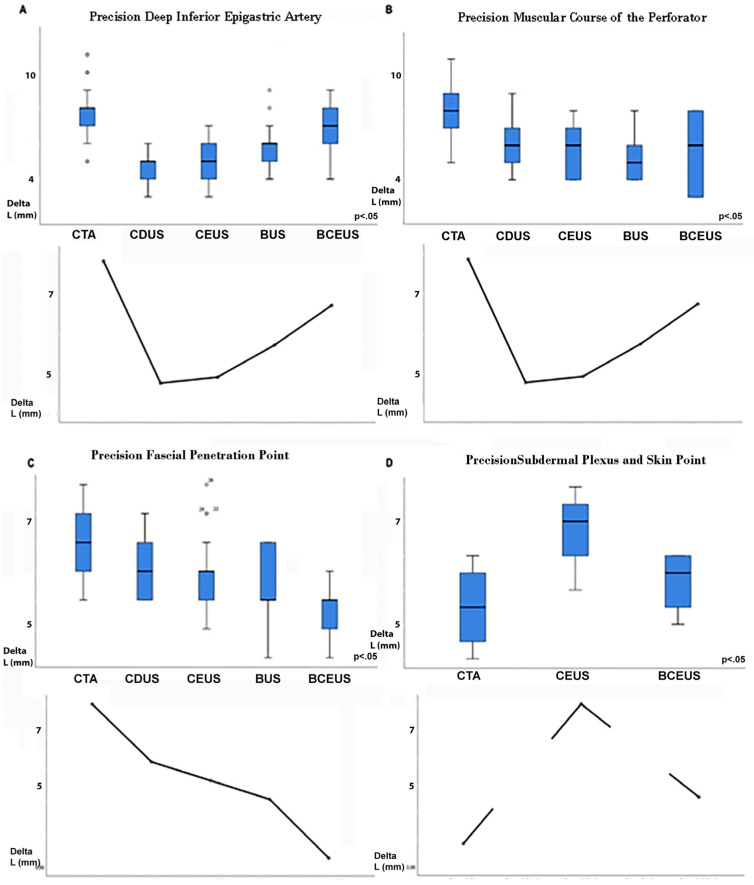
**Figure 3** displays the precision of PM, measured as ∆L (mm), corresponding to the difference between preoperative PM and the actual intraoperative situation. (**A**) shows the precision of DIEA PM for each imaging mode. (**B**) displays the precision of M, while (**C**) displays the precision of F for each mode. (**D**) clearly shows that only CTA and BCEUS were able to reliably display P and SP.

**Figure 4 jpm-13-00064-f004:**
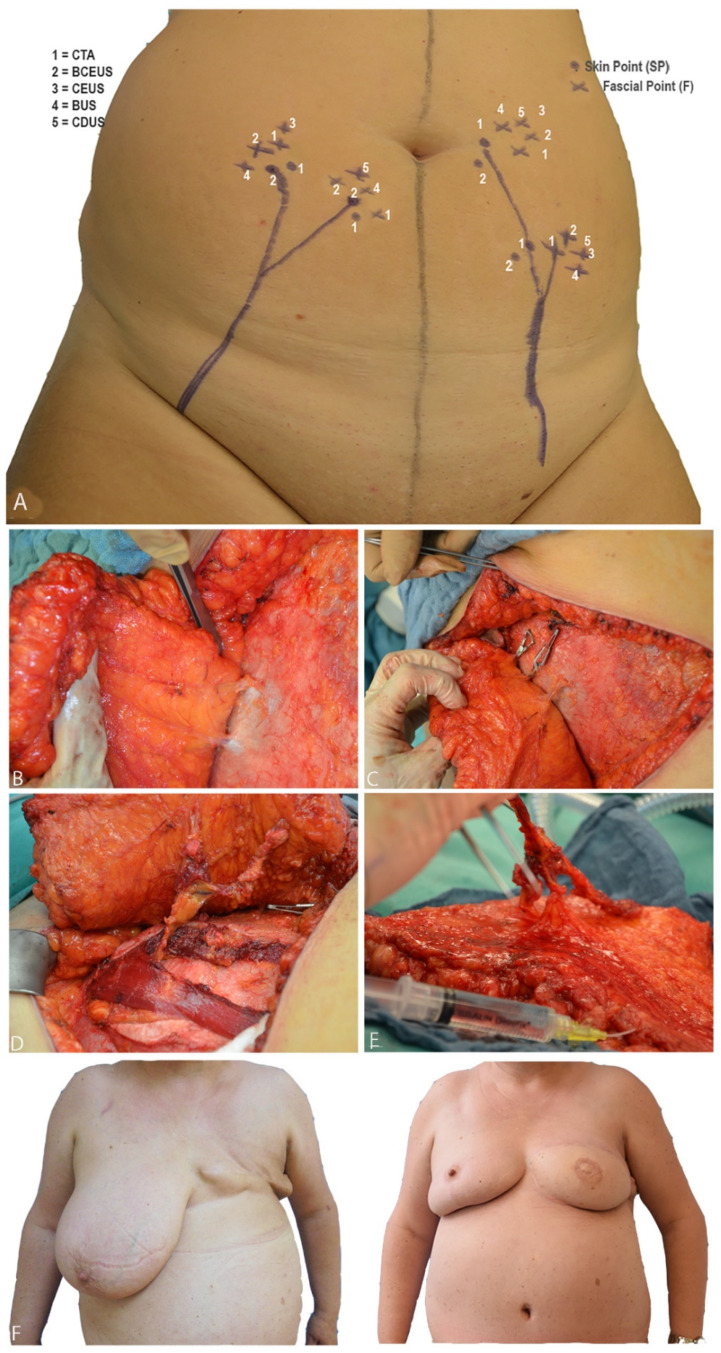
(**A**) shows preoperative PM. When BCEUS, BUS, CEUS and CDUS were used, landmarks could be directly navigated on the abdomen. (**B**) illustrates F and P of the respective perforators. In (**C**), the clamping test is shown. (**D**) portrays the intramuscular dissection of the perforator vessels down to the origin of the DIEA. (**E**) shows the harvested fasciocutaneous DIEP flap, including the perforator vessels. (**F**) shows pre- and postoperative pictures of a patient after breast amputation and consecutive DIEP flap reconstruction, including contralateral breast reduction.

**Table 1 jpm-13-00064-t001:** Patient data.

**No. of Flaps** **No. of Patients**	**39** **30**
**Mean BMI (kg/m^2^)**	24.7 SD ± 3.9
**Mean age (years)**	52 SD ± 46
**Hemi-abdomen analyzed**	60
**Autologous breast reconstructions**	
**Single sided**	21
**Double sided**	9
**DIEA branching pattern** (Type I-III, Moon, Taylor [20]) **Type I** 22% **Type II** 49% **Type III** 29%	

BMI: body mass index in kg/m^2^. SD: standard deviation.

**Table 2 jpm-13-00064-t002:** Technical details CT scans and ultrasound examinations.

CT Scan Parameters	
**Scanner**	IQon (Philips Healthcare, Best, The Netherlands)
**Slice thickness**	2 mm
**Detector pitch**	0.671
**Gantry rotation time**	0.5 s
**Tube potential**	120 kV
**Tube current**	Tube current modulation activated by default (DoseRight 3D-DOM; Philips Healthcare, Best, The Netherlands).
**IV contrast**	Accupaque 350 mg/mL (GE Healthcare; Little Chalfort, UK); IVI 4 mL/s
**Range**	Xiphisternum to the pubic symphysis
**Bolus tracking**	150 HU within the abdominal aorta (activated bolus tracking) Cranio-caudal direction, supine position and inspirational breath hold
**Ultrasound parameters**	
**Scanner**	LOGIQ E9 (GE Healthcare, Chicago, IL, USA)
**Linear probe**	9L-D; 2–8 MHz (GE Healthcare, Chicago, IL, USA)
**Matrix probe**	ML6-15; 4–15 MHz (GE Healthcare, Chicago, IL, USA)
**Modes**	Color Doppler ultrasound (CDUS) Contrast-enhanced ultrasound (CEUS) B-flow imaging (BUS) Contrast-enhanced B-flow imaging (BCEUS)
**IV contrast**	SonoVue 2 mL (Bracco^®^, Milan, Italy)

CT (computed tomography), IV (intravenous), IVI (intravenous injection), HU (Hounsfield units).

**Table 3 jpm-13-00064-t003:** Precision of preoperative perforator mapping.

Perforator Mapping	Diagnostic Mode				
	**CTA**	**CDUS**	**CEUS**	**BUS**	**BCEUS**
**Deep inferior epigastric art.**					
**Mean ∆L (mm)**	7.85	4.77	4.92	5.73	6.73
**Deviation ± (mm)**	±1.57	±0.863	±1.02	±1.31	±1.43
**Confidence interval (95%)**	7.21–8.48	4.42–5.12	4.51–5.33	5.20–6.26	6.15–7.31
**Kruskall–Wallis**	*p* < 0.05 significance between the groups				
**Levene test**	*p* > 0.05 equality of variances				
**Post hoc-test (Bonferroni)**	Reference	*p* < 0.05	*p* < 0.05	*p* < 0.05	*p* = 0.19
**Correlation Spearman**	Reference	R = 0.372 (*p* < 0.05)	R = −0.10 (*p* = 0.31)	R = 0.19 (*p* = 0.17)	R = 0.09 (*p* = 0.33)
** *Muscular course (M)* **					
**Mean ∆L (mm)**	8.04	5.92	5.88	5.50	5.42
**Deviation ± (mm)**	±1.59	±1.32	±1.56	±1.34	±2.18
**Confidence interval (95%)**	7.39–8.67	5.39–6.46	5.26–6.51	4.96–6.04	4.54–6.30
**Kruskall–Wallis**	*p* < 0.05 significance between the groups				
**Levene test**	*p* > 0.05 equality of variances				
**Post hoc-test (Bonferroni)**	Reference	*p* < 0.05	*p* < 0.05	*p* < 0.05	*p* = 0.19
**Correlation Spearman**	Reference	R = −0.233 (*p* = 0.126)	R = 0.131 (*p* = 0.261)	R = −0.172 (*p* = 0.201)	R = −0.216 (*p* = 0.144)
** *Fascial penetration point (F)* **					
**Mean ∆L (mm)**	6.00	5.12	4.83	4.54	3.63
**Deviation ± (mm)**	±1.17	±1.03	±1.19	±1.21	±0.84
**Confidence interval (95%)**	5.53–6.47	4.70–5.54	4.35–5.31	4.05–5.03	3.29–3.98
**Kruskall–Wallis**	*p* < 0.05 significance between the groups				
**Levene test**	*p* > 0.05 equality of variances				
**Post hoc-test (Bonferroni)**	Reference	*p* < 0.05	*p* < 0.05	*p* < 0.05	*p* < 0.05
**Correlation Spearman**	Reference	R = −0.361 (*p* < 0.05)	R = 0.116 (*p* = 0.126)	R = −0.059 (*p* = 0.387)	R = 0.799 (*p* < 0.05)
**Subcutaneous course of the plexus (P) and skin point (SP)**					
**Mean ∆L (mm)**	2.85	no signal	8.00	no signal	4.57
**Deviation ± (mm)**	±1.26	--	±0.82	--	±1.34
**Confidence interval (95%)**	2.34–3.35	--	6.70–9.30	--	3.98–5.15
**Kruskall–Wallis**	*p* < 0.05 significance between the groups				
**Levene test**	*p* > 0.05 equality of variances				
**Post hoc-test (Bonferroni)**	Reference	--	*p* < 0.05	--	*p* = 0.32
**Correlation Spearman**		--	R = −0.316 (*p*=0.342)	--	R = 0.778 (*p* < 0.05)

∆L = intraoperative situation—preoperative diagnostic planning.

**Table 4 jpm-13-00064-t004:** Number of perforators at the fascial level F.

Number of Perforators (F)	Intraoperative Situation	Diagnostic Modes				
	Real	CTA	CDUS	CEUS	BUS	BCEUS
**Mean**	2.59	2.48	1.10	1.40	1.81	2.27
**Deviation ±**	±0.129	±0.122	±0.069	±0.072	±0.076	±0.103
**Sum**	101	97	43	55	70	88.5
**Deviation ±**	±1.487	±1.411	±0.803	±0.827	±0.874	±1.185
**Correlation Spearman**	Reference (1.00)	0.981	0.091	0.225	0.584	0.947
**Significance**		*p* = 0.346	(*p* < 0.05)	(*p* < 0.05)	(*p* < 0.05)	*p* = 0.173

**Table 5 jpm-13-00064-t005:** Delineation of perforator mapping and diagnostic efficiency.

Delineation of Perforator Mapping	Diagnostic Modes				
	CTA	CDUS	CEUS	BUS	BCEUS
**Deep inferior epigastric artery (DIEA)**	Reference				
Sensitivity (%)	--	100	100	100	100
Specificity (%)	--	100	100	100	100
Youden index (%)	--	100	100	100	100
**Muscular course (M)**	Reference				
Sensitivity (%)		91.3	95.7	96.3	92
Specificity (%)	--	95.4	66.7	66.7	69.4
Youden index (%)	--	86.7	62.4	63	61.4
**Fascial penetration point (F)**	Reference				
Sensitivity (%)		68.2	81.8	72.7	90.9
Specificity (%)	--	93	92	94	92
Youden index (%)	--	61.2	73.8	66.7	82.9
**Subcutaneous course (P)**	Reference				
Sensitivity (%)		38.1	71.4	52.4	90.1
Specificity (%)	--	68.6	75.3	70.4	93.2
Youden index (%)	--	6.7	46.7	22.8	83.3
**Skin point (SP)**	Reference				
Sensitivity (%)	--	15.8	52.6	42.1	91.2
Specificity (%)	--	72.4	57.1	68.3	88.9
Youden index (%)	--	−11.8	9.7	10.4	80.1

**Table 6 jpm-13-00064-t006:** Recommended ultrasound ModeAlgorithm for DIEP perforator mapping.

Ultrasound Mode	Perforator Mapping
**Color Doppler ultrasound** **(CDUS)**	Inferior epigastric artery (IEA) Muscular course (M)
**Contrast-enhanced B-flow imaging** **(BCEUS)**	Fascial point (F) Subcutaneous plexus (P) Skin point (SP)

## Data Availability

Not applicable.

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
