# Peer review of "Preoperative Perforator Mapping in DIEP Flaps for Breast Reconstruction. The Impact of New Contrast-Enhanced Ultrasound Techniques"

_jpm, 2022, doi:10.3390/jpm13010064_

Round 1
Reviewer 1 Report
I congratulate with the Authors for that wonderful work. I suggest publication for the high impact and originality.
Author Response
Thank you very much for this kind comment.
Reviewer 2 Report
The authors present a manuscript on the comparison of ultrasound-based modalities for the perforator mapping in the context of autologous breast reconstruction. In their manuscript they also investigated the precision and the effectiveness of the modalities and came to the result that BCEUS even presents an “equivalent alternative to CTA”. Unfortunately, the calculation and measurements of the metric deviation and the comparison itself are not presented in full detail. Therefore, a revision of the manuscript is necessary in which the authors edit the manuscript and present more detail on the calculation itself as well as the exact measurement techniques.
Also, it would be nice if the authors would present an estimation to what extent the new technique will find or has already been introduced in their routine perforator mapping in their different breast centers / departments and clincs.
Furthermore, the manuscript needs editing of English language and style.
Line 2: Please capitalize the acronym DIEP
Line 6 ff: Please list author affiliations according to position in the authorship, add Christoph Andree C‘s affiliation
Line 25: Please write out the abbreviation DIEP at first appearance
Line 27: instead of “centrally located” rather state “sufficient”
Line 28: “We therefore investigated” ïƒ Therefore, we investigated
Line 29: “compaired” ïƒ compared
Line 31: add “vascular” before pedicle
Line 37: Please present the exact value for ΔL in CTA (2,85) as well and not only for BCEUS (4,57)
Line 52: „of“ ïƒ for autologous breast reconstruction
Line 53 ff: Please add the references for all the listed features and correct the citation as Gill et al. did not investigate the hospital stay themselves but rather cited the manuscript by “Kaplan, J. L., and Allen, R. J. Cost based comparison between perforator flaps and TRAM flaps for breast reconstruction. Plast. Reconstr. Surg. 105: 943, 2000.”
Line 62: Please add a reference here
Line 70: Please change “…” to (e.g. velocity, flow)
Line 71: Instead of “calculated” rather say referred to the umbilicus unless the authors really use mathematical calculations
Line 89f: Please delete this sentence here and rather add it in the discussion section.
Line 96: Please change “We thus hypothesized” ïƒ Thus, we hypothesized …
Line 102: Please add the number of patients in which the autologous breast reconstruction was performed. Mismatch to Table 1: patients 39, flaps 30
Line 104: Please add the referral code of the University ethical committee either in Section 2 or at the end of the manuscript “Institutional Review Board Statement”
Line 110: In table 1 the DIEA branching pattern is listed, here the DIEP branching pattern is named, please correct accordingly.
Line 130: Please delete “normal”
Line 131: How many investigators are represented in “we”, and what is meant by “distinct” what is distinct enough?
Line 149f: Please explain in more detail what kind of comparison was performed. The actual statement is not clear on the identification in the sense of location, course of the perforators, general identification or in the exact comparison to interoperative findings. Please add details on the measurement techniques also used for CTA PM.
Line 150: Please present more detail on how metric deviation was analyzed
Line 158: Please add company details for SPSS
Line 167ff: In table 1 the number, if the table presents the total number (n=) in the section “Autologous breast reconstruction”, there are 18 double side autologous reconstructions listed, please correct the number either in the manuscript or the table to the correct number. Furthermore, please add the unit for BMI in the manuscript as well as in the table.
Line 195 f: The presentation of the topics background is not necessary to state in the results section, please edit it accordingly
Line 201: Please add more detail in what sense BCEUS and CEUS are feasible alternatives compared to CTA, does this refer to the aforementioned facts?
Line 208: Please rephrase the statement because with the used wording it is not exactly clarified if CDUS is recommended for identification of the DIEA or the intramuscular course or both, as later on presented in table 6.
Line 212: Please add the mean duration of CT scans
Line 222: Please add as stated in the results section that the values in the table refer to the identification in the epifascial level (F).
Line 224f: What are the abbreviations and their explanations refer to, as there is e.g. no Perf. Statement in the table?
Line 226: Please add in the table subscription “Recommended ultrasound mode algorithm…”
Line 233: For Figure 2 please edit the figure for a better identification of especially the intramuscular course and the single layers in the ultrasound images.
Line 238: Figure 3 needs to present in a better detail and resolution because it is not understandable what data is presented in the graphs for A-D. The histograms should give detail on found significances.
Line 245: Please present Figure 1 A in a better and bigger resolution, as I understand this figure as the most important figure because it illustrates the topic of the manuscript. Furthermore, I would suggest erasing figure 4 D and 4 E in this multi-image figure although it presents how nicely you dissected the vascular course but doesn’t give any detail on the topic.
Line 272: the term efficiency can be misunderstood as the reader might think about the duration of the investigation. Therefore, please clarify that here only the efficiency in concern to the identification is referred to.
Line 273: please add the compared technique to what BCEUS “is more precise and advantageous”.
Line 281: the authors probably refer to CDUS, please correct the abbreviation
Line 298: Please correct This / thus
Line 300: As 3D reconstruction/volume rendering is discussed it would be nice to get a statement by the authors to what extend the 3D reconstruction/volume rendering is valued in the context of CTA being the gold standard.
Line 322: In the statement on cost efficiency, it is not clear if the authors refer to financial costs or the cost of radiation to the patient. If financial cost is referred to the authors should present more details and references.
Line 347: Although the manuscript is on the comparison of the different perforator mapping techniques, it would be interesting as the authors state that that BCEUS is an “equivalent alternative” to get in information on how many patients have been operated on, in which only BCEUS perforator mapping was performed, and CTA was not performed. To put it straight: are the authors themselves convinced enough that they skip CTA and only relay on BCEUS ultrasound perforator mapping in the future.
Author Response
Please see the attachment. Word document.
Author respond to reviewer.

Reviewer 3 Report
First of all, I would like to congratulate the authors who have come with such an interesting topic to level up the usability of ultrasound; in this case is in microsurgery field (DIEAP flap for breast reconstruction).
Although it is clear enough to understand the methodology, however, it is not clear on how the authors could describe their own bias when they made decision on table whether these CTA made a strong influence compared to the decision which was merely based on the ultrasound (BCEUS+CDUS or others) mapping?
It is strongly advised that the authors identify their study design in the Methods. When they think their study a descriptive anatomical study, for example, then it would be better they declare it. It is also necessary to allow the future readers know about the golden standard whether it is the CTA or the intra-operative findings.
Every subjects underwent multimodal examination, which might not be considered a standard practice, provided no informed consent (on the excessive examination process). In fact the practice was then made as part of study, even though retrospective study. Please explain this and put the notes in the manuscript.
The authors need to describe on how the combined ultrasound modes would be impacting the clinical outcome.
Last but not least, the tittle is too long. It is suggested that the authors revise it to become shorter, yet a clear and brief title.
Author Response
Please see the attachement.
Author respond to reviewer 3.
